# Speckle Tracking-Derived Left Atrial Stiffness Predicts Clinical Outcome in Heart Failure Patients with Reduced to Mid-Range Ejection Fraction

**DOI:** 10.3390/jcm9051244

**Published:** 2020-04-25

**Authors:** Ibadete Bytyçi, Frank L. Dini, Artan Bajraktari, Nicola Riccardo Pugliese, Andreina D’Agostino, Gani Bajraktari, Per Lindqvist, Michael Y. Henein

**Affiliations:** 1Institute of Public Health and Clinical Medicine, Umeå University, 90187 Umeå, Sweden; i.bytyci@hotmail.com (I.B.); artan.bajraktari@umu.se (A.B.); ganibajraktari@yahoo.co.uk (G.B.); per.lindqvist@umu.se (P.L.); 2Clinic of Cardiology, University Clinical Centre of Kosovo, 10000 Prishtina, Kosovo; 3Cardiac, Thoracic and Vascular Department, University of Pisa, 56124 Pisa, Italy; franklloyddini@gmail.com (F.L.D.); n.r.pugliese88@gmail.com (N.R.P.); andreina_dagostino@libero.it (A.D.); 4Faculty of Medicine, Department of Surgical and Perioperative Sciences, Clinical Physiology, Umeå University, 90187 Umeå, Sweden; 5Molecular and Clinic Research Institute, St George University, London SW17 0QT, UK; 6Institute of Fluid Dynamics, Brunel University, London UB8 3PH, UK

**Keywords:** heart failure, cardiac events, left atrial stiffness, clinical outcome

## Abstract

Background and Aim: Left atrial stiffness (LASt) is an important marker of cardiac function, especially in patients with heart failure (HF); it explains symptoms on the basis of pressure transfer to the pulmonary circulation. The aim of this study was to evaluate the relationship between LASt and cardiac events (CE) in HF patients with reduced to mid-range ejection fraction. Methods: The study included 215 consecutive ambulatory HF patients with ejection fraction (EF) < 50% (162 HF reduced EF and 53 HF mid-range EF) of mean age 66 ± 11 years and 24.4% females. Peak LA strain (PALS) was measured by speckle tracking echocardiography and E/e’ recorded from the apical four-chamber view. Non-invasive LASt was calculated using the equation: LASt = E/e’ ratio/PALS. Documented cardiac events (CE) were HF hospitalization and cardiac death. Results: During a median follow up of 41 ± 34 months, 65 patients (30%) had CE. In multivariate analysis model, only raised LV filling pressure (E/e’) (OR = 0.292, (95% CI 0.099 to 0.859), *p* = 0.02), peak pulmonary artery pressure (PAP) (OR = 1.050 (1.009 to 1.094), *p* = 0.01), PALS (OR = 0.932 (0.873 to 0.994), *p* = 0.02) and LASt (OR = 3.781 (1.144 to 5.122), *p* = 0.001) independently predicted CE. LASt ≥ 0.76% was the most powerful predictor of CE, with 80% sensitivity and 73% specificity (AUC 0.82, CI = 0.73 to 0.87, *p* < 0.001) followed by PALS ≤ 16%, with 74% sensitivity and 72% specificity (AUC 0.77, CI = 0.71 to 0.84, *p* < 0.001). These results were consistent irrespective of EF (*p* < 0.05). Conclusion: In this cohort of ambulatory HFrEF and HFmrEF patients, LASt proved the most powerful predictor of clinical outcome.

## 1. Introduction

Heart failure (HF) is becoming a major public health problem worldwide due to its increasing prevalence, especially in developed countries [1]. Despite introducing new pharmacological agents and various devices that have markedly improved clinical outcome including prolonged life expectancy, disease-related mortality remains unacceptably high [2]. The current European Society of Cardiology (ESC) guidelines reconsidered the classification of HF and three types have now been defined: HFpEF (Left ventricle (LV) ejection fraction (EF) ≥ 50%), HFmrEF (LVEF 40–49%) and HFrEF (LVEF < 40%) [3]. However, the treatment of the two types of HF patients with EF < 50% remains similar [4,5].

Over the last decade, there has been increasing recognition of the importance of left atrial (LA) structure and function in the pathophysiology of HF [6,7,8]. The cardiac pathology of raised LA pressure resulting from decreased LVEF is well established, despite its limited routine assessment in daily clinical practice [9]. Studies have shown that raised LA pressure is secondary to increased LA cavity stiffness which results from diffuse myocardial fibrosis [10]. Although traditional assessment of LA pressure and cavity stiffness is invasively achieved, available Doppler echocardiographic techniques, particularly atrial deformation and myocardial velocities, have proved accurate in providing such important LA function information but prognostic application of those measurements in HF patients remains lacking [11].

The aim of this study was to evaluate the relationship between non-invasive LA stiffness measurements and cardiac events (CE) in HF patients with reduced to mid-range ejection fraction.

## 2. Methods

### 2.1. Study Population

We studied 215 consecutive patients with clinical signs and symptoms of HF (New York Heart Association—NYHA class I–III), and LV EF < 50%, (162 HFrEF and 53 HFmrEF) according to the current ESC guidelines [3]. Patients were referred to the Cardiac, Thoracic and Vascular Department, University of Pisa, Pisa, Italy between January 2013 and December 2018.

Exclusion criteria were: atrial arrhythmia, history of congenital heart disease, pacemaker insertion, valve surgery, cardiac transplantation, chronic obstructive pulmonary disease (COPD) or recent acute coronary syndrome, stroke, poor echocardiographic window and the age < 18 years. The study was approved by the local institutional review board (20110015213) and all patients gave written informed consent before enrollment in the study. The study was conducted in accordance with institutional policies, national legal requirements, and the revised Helsinki Declaration.

### 2.2. Data Collection

Detailed history and clinical assessment were obtained in all patients, in whom routine blood tests were performed, including complete blood count, blood glucose, kidney function tests. Weight and height measurements were also obtained in all study patients.

### 2.3. Echocardiographic Examination

All echocardiographic examinations were made by one experienced sonographer, using an iE33 X5-matrix Ultrasound instrument (Philips, Andover, MA, USA) equipped with a 2.5–3.5 MHz phased-array sector scan probe and second harmonic technology. LV end-systolic and end-diastolic dimension measurements were made from the left parasternal long axis view with the M-mode cursor positioned by the tips of the mitral valve leaflets. LV volumes and EF were calculated from the apical two and four chamber views using the modified Simpson’s method [12]. Right ventricular (RV) global systolic function was assessed from tricuspid annular plane systolic excursion (TAPSE) of the lateral tricuspid annulus angle (in cm), in the apical four-chamber view and RV long axis myocardial velocities were obtained using Doppler myocardial imaging technique and conventional protocols [13]. Peak pulmonary artery systolic pressure (PAP) was estimated from transtricuspid retrograde pressure drop (from tricuspid regurgitation (TR) peak velocity) using continuous wave (CW) Doppler and applying the simplified Bernoulli equation (P= 4 × TRmax^2^) [14]. The E/A ratio represents the ratio of peak left ventricle filling velocity in early diastole (E wave) to that in late diastole, during atrial contraction (A wave). LV E/e’ ratio was calculated as the ratio between peak trans-mitral E wave velocity and mean lateral and septal LV e’ wave velocities. An E/e’ ratio cut off of ≥15 was used as a marker of raised LV filling pressure [15]. Th markers of LV restrictive filling pattern were E/A ratio >2, short E deceleration time (<140 ms) and reduced peak early diastolic mitral annular velocity (e’) [16]. Mitral regurgitation severity was assessed using color, continuous-wave Doppler and other conventional quantitative parameters including the relative mitral regurgitation jet area to that of the LA. Mitral regurgitation velocity profile was graded as mild, moderate, or severe, according to the guidelines of the American Society of Echocardiography [17]. 

### 2.4. Measurements of LA Structure and Function

LA diameter was measured in the parasternal long axis view (anterior–posterior diameter) from aortic root recordings with the M-mode cursor positioned at the level of the aortic valve leaflets and from the 4-chamber view (longitudinal and transverse diameters). LA volumes were measured using area-length method from the apical four and two chamber views, in line with the recommendations of the American Society of Echocardiography [18]. LA size was measured at the LV end-systole when the LA chamber was at its largest volume, in the long-axis view (anterior–posterior diameter) and 4-chamber view (longitudinal and transverse diameters). LA maximum volume (LAV max) was measured at LV end-systole, just before the mitral valve opening. LA minimum volume (LAV min) was measured at end-diastole, directly after mitral valve closure, and LA maximum volume index (LAVI max) was defined as LAV max divided by body surface area. Likewise, LAV min was measured and LAVI min was calculated [19]. 

Global systolic LA myocardial strain was measured using 2-dimensional speckle tracking echocardiography technique [20]. Grey scale imaging of the apical 4-chamber views was obtained with frame rates of 50–80 fram/s. Recordings were processed with acoustic tracking software (Cardiac Motion/Mechanic Quantification [Q-Lab] 10, Philips Medical, Andover, Massachusetts), allowing off-line semi-automated speckle-based strain analysis. LA inner myocardial border was manually traced at the LV end-systole. An epicardial border tracing was then automatically generated by the software, which created a region of interest (ROI). After manually adjusting the ROI shape and position, peak LA strain (PALS) during the whole cardiac cycle was estimated, and LASt was calculated using the formula (Figure A1 in Appendix A) [21,22,23]: LASt = E/e’/PALS(1)

### 2.5. Follow up

Cardiovascular events (CE) were prospectively reported during follow-up. Information on patients’ clinical outcome was obtained through clinical visits, personal communication with general physicians, and telephone interviews with patients and relatives, with the help of trained research nurses. The primary study end points were CE, combined death and hospitalization for worsening HF and secondary end points were all-cause mortality, cardiac death and HF-related hospitalization. 

### 2.6. Statistical Analysis

Data are summarized using frequencies (percentages) for categorical variables and mean ± standard deviation for continuous variables or median interquartile (IRQ) ranges, when appropriate continuous data were compared with two-tailed Student *t*-test and discrete data with Chi-square test. Correlations were tested using Pearson coefficients. Predictors of cardiac events were identified with univariate analysis and multivariate logistic regression which was performed using the step-wise method. CART analysis was used to generate the best decision tree in predicting CE. The following predictors of CE were considered in the analysis: LASt ≥ 0.99%, E/e’ ≥ 13 OR PAPs > 35 mmHg and LV filling pattern (defined as DT < 120 ms and E/A ratio > 2). The diagnostic accuracy of the CART algorithm was compared with that of individual parameters using the exact binomial test. The relative increases in accuracy observed in patients with HFrEF and HFmrEF were compared using the Fisher exact test. 

The receiver operator characteristic (ROC) analyses were performed and the best cut-off values with its sensitivity and specificity were determined. A significant difference was defined as *p* value < 0.05 (2-tailed). Statistical analysis was performed using SPSS Software (IBM Corp., Armonk, NY, USA) Package version 22.0.

## 3. Results

### 3.1. Clinical, Demographic and Echocardiographic Data 

Patients’ mean age was 66 ± 11 years and 24.4% were females. All patients were in sinus rhythm, had EF < 50% and NYHA class ≤ 3, 162 (75.3%) had EF < 40% (HFrEF) and the remaining 53 (22.6%) had LVEF 40–49% (HFmrEF). A total of 73.6% of patients had raised BNP > 125pg/mL, 40.6% were hypertensives, 31.3% diabetics, 26.3% had chronic renal failure and 48.8% had coronary artery disease. 80.5% of the study patients were taking diuretics, 87% beta-blockers and 86.9% ACE-I or ARBs. A total of 71 (33%) patients had E/e’ > 15, 15.9% had a restrictive LV filling pattern and insignificant (less than moderate) mitral regurgitation was present in 90 patients (43%) (Table A1).

### 3.2. Clinical and Demographic Data of Patients with Vs without CE

At a mean follow up of 41.4 ± 34 months, 63 (29.3%) patients had CE and the remaining 152 (70.6%) had no CE. Patients with CE were in a higher NYHA class (*p* < 0.001) and had a higher BNP level (*p* = 0.001), but age and gender were not different (*p* > 0.05 for both) from those without CE. They also had more comorbidities: systemic hypertension (*p* = 0.01), diabetes (*p* = 0.01), chronic renal failure (*p* = 0.001) and coronary artery disease (*p* = 0.01). They received more diuretics (*p* < 0.001) but beta- blockers, ACE-I and/or A RBs (*p* > 0.05 for both) were not different between groups (Table 1). 

### 3.3. Cardiac Function in Patients with Vs without CE

Patients with CE had larger LV dimensions (LVEDD (*p* = 0.01) and LVESD (*p* < 0.001)), larger LV volumes (LVEDV (*p* = 0.01) and LVESV (*p* = 0.009)), lower EF (*p* = 0.001), higher E/A ratio (*p* = 0.04) and frequent insignificant (less than moderate) mitral regurgitation (*p* < 0.001) compared to those without CE. In addition, PAP was higher (*p* = 0.01) and RV longitudinal systolic function lower (*p* < 0.001) in patients with CE. They also had larger LA dimensions (*p* < 0.001) and LA volumes (LAVI max and LAVI min, *p* < 0.001 for both), lower PALS (*p* < 0.001) and higher LA stiffness (*p* < 0.001), compared with those without CE (Table 1). 

### 3.4. Predictors of CE

In univariate analysis, LVEF (*p* < 0.001), NYHA class (*p* < 0.001), PSAP (<0.001), LV E/e’ ratio (*p* < 0.001), TAPSE (*p* = 0.001) and LA structure and function (LA diameter (*p* = 0.001), LAVI max (*p* < 0.001), LAVI min (*p* < 0.001), PALS (*p* < 0.001) and LASt (*p* < 0.001)) predicted CE.

In the multivariate analysis model, only increased LV filling pressure: E/e’>15 (OR = 0.292, (95% CI 0.099–0.859), *p* = 0.02), PAP (OR =1.050 (1.009–1.094), *p* = 0.01) PALS (OR = 0.932 (0.873 to 0.994), *p* = 0.02) and LASt (OR = 3.781 (1.144 to 5.122), *p* = 0.001) independently predicted CE (Table 2). Collinearity between these measurements was not met based on VIF < 10 for all predictors. A LASt ≥ 0.76% predicted CE with 80% sensitivity and 73% specificity (AUC 0.82, CI = 0.73–0.87, *p* < 0.001), predicted cardiac death with 86% sensitivity and 64% specificity (AUC 0.80, CI = 0.66–0.88, *p* < 0.001), all-cause mortality with 76% sensitivity and 64% specificity (AUC 0.76, CI = 0.66–0.87, *p* < 0.001) and HF related hospitalization (AUC 0.72, CI = 0.63–0.84, *p* < 0.001) with 75% sensitivity and 64% specificity (Figure 1a–d).

A PALS ≤ 16% predicted CE with the following accuracies: overall CE 74% sensitivity and 72% specificity (AUC 0.77, CI = 0.70–0.84, *p* < 0.001), cardiac death with 79% sensitivity and 64% specificity (AUC 0.77, CI = 0.62–0.81, *p* < 0.001), all-cause mortality with 78% sensitivity and 64% specificity (AUC 0.76, CI = 0.67–0.85, *p* < 0.001), and HF hospitalization (AUC 0.68, CI = 0.66–0.81, *p* = 0.001) with 69% sensitivity and 63% specificity (Figure 2a–d). 

A PAP ≥ 36 mmHg predicted CE with 75% sensitivity and 70% specificity (AUC 0.75, CI = 0.69–0.83, *p* < 0.001), cardiac death with 76% sensitivity and 64% specificity (AUC 0.76, CI = 0.68–0.84, *p* < 0.001), all-cause mortality with 75% sensitivity and 64% specificity (AUC 0.74, CI = 0.66–0.82, *p* < 0.001) and HF-related hospitalization (AUC 0.66, CI = 0.58–0.75, *p* < 0.001) with 69% sensitivity and 62% specificity (Figure A2a–d).

### 3.5. Predictors of CE according to EF

A LA stiffness ≥0.76% was 75% sensitive and 70% specific (AUC 0.79, CI = 0.72–0.86, *p* < 0.001) in predicting CE in patients with HFrEF and 75% sensitive and 90% specific (AUC 0.84, CI = 0.99–0.86, *p* = 0.003) in patients with HFmrEF (Figure 3a,c). PALS ≤ 16% had similar predictive powers for CE with 78% sensitivity and 68% specificity (AUC 0.77, CI = 0.69–0.85, *p* < 0.001) in HFrEF patients and 65% sensitivity and 80% specificity (AUC 0.75, CI = 0.54–0.96, *p* = 0.02) in those with HFmrEF (Figure 3b,d).

### 3.6. Classification and Regression Tree Analyses 

The rank order of single or combined echo parameters to predict CE was derived using the statistical method. The LASt ≥ 99%was detected as the root node which identified 55 out of 78 with 70% sensitivity, 91% specificity and 85% accuracy. E/e’ ≥ 13 OR PAPs > 35 mmHg was generated as second decision node with 60% sensitivity, 96% specificity and 82% accuracy. Patients who did not fulfil this criterion were considered without CE, and no further criteria were needed. For the remaining patients, CART analysis generated a third complex node that made DT <120 and E/A >2 predict CE which identified five out of nine with 52% sensitivity, 90% specificity and 81% accuracy (Figure 4, Table 3). CART sub-analysis was also performed in the two groups of patients according to ejection fraction: HFrEF and HFmrEF. In HFrEF patients, the first node included LASt ≥ 99%, which identified 48/69 patients with CE and 73/95 without CE, with 86% sensitivity, 78% specificity and 82% accuracy. The second node was DT <120 and E/A >2, with 71% sensitivity, 50% specificity and 56% accuracy. Finally, the E/e’ ≥ 13 OR PAPs > 35 mmHg identified 5/7 with CE and all three patients without CE, with 58% and 100% specificity and 70% accuracy. In a similar way, the first node in HFmrEF was LASt ≥ 99%, which identified 5/9 patients with CE with 57% sensitivity, 84% specificity and 79% accuracy. The second node was DT <120 and E/A >2 with 52% sensitivity and 84% specificity. In addition, the E/e’ ≥ 13 OR PAPs > 35 mmHg as third node corrected the diagnostic accuracy having identified 9/15 patients with CE and 23/28 without CE, with 65% sensitivity, 82% specificity and 70% accuracy (Figure 4, Table 3).

## 4. Discussion

**Findings:** The main findings in this study are summarized as follows: (1) LA stiffness, PALS and NYHA class predicted cardiac events in HF patients with reduced and mid-range EF; (2) LA stiffness was superior in predicting cardiac events in these HF patients, irrespective of EF; (3) Patients with cardiac events had increased LA stiffness, LV dimensions and its function was worse than in those with no events; (4) LA stiffness was related to raised LA pressure and the two were related to the occurrence of clinical cardiac events.

**Data interpretation:** Our findings show that compromised LA function in the form of reduced myocardial strain and increased LA stiffness is associated with signs of raised cavity pressures, estimated non-invasively using Doppler markers. These disturbances in LA structure and function proved the main discriminatory factors between patients who developed cardiac events and those who remained stable. In the same group of patients with higher LA stiffness, the LV was larger with worse systolic function. Such findings should not be taken in isolation [24]. Irrespective of the underlying pathology, the phenotypic severity of LV dysfunction seems to be the main association with worse LA function and stiffness [21,25,26]. The strong relationship we found between worse LA function and clinical events is of significant importance and needs to be thoroughly addressed for potential clinical benefit. Impaired LV systolic function is commonly associated with incompliant cavity. Limited increases in cavity size in diastole are associated with raised diastolic pressures which chronically result in the perpetual rise of LA pressures, increase in cavity volume and myocardial stretch [27,28]. These structural changes are bound to eventually compromise LA’s intrinsic myocardial properties, reduce cavity deformation and increase its stiffness and fibrosis [28,29,30]. The end result of such abnormalities is the dilatation of mitral annulus, worsening mitral regurgitation, pulmonary venous hypertension and, later on, pulmonary arterial hypertension [31]. Again, with long-standing myocardial stiffness, the condition becomes irreversible, with worsening symptoms, secondary implications on right heart structure and function and deterioration of overall cardiac performance, reduced stroke volume and cardiac output, with limited benefit from medical treatment [32,33]. Our results summarize such a pathophysiological scenario and its relationship with clinical outcome in the way of repeated HF-related hospitalization and increased mortality. They also show how individual markers of cardiac chamber function are related and become more dependent on each other in severe disease, in order to maintain a satisfactory stroke volume [34].

**Limitations:** The relatively small number of patients is an important limitation of this study. The heterogenous underlying pathology limited the classification of patients into subgroups for further comparisons. LV deformation assessment was not available, which could have added an important arm in the analysis and prediction of clinical outcome. We are not sure about potential relationships between the medications not shared between all patients and their potential impact on the results, since diuretics were more commonly used in patients who developed clinical events, a finding that suggests that stiffer myocardium may cause more congestion. Our results are likely to have limited application in patients with atrial fibrillation, since all our patients were in sinus rhythm. Likewise, the relevance of our findings in HF patients with significant mitral regurgitation needs to be retested, since our findings concern raised LA pressure due to pressure factors rather than volume impact. Equally important is the potential application of our findings in patients with HFpEF remains to be confirmed. Although in the multivariate analysis model LA stiffness evolved as the most powerful predictor of clinical outcome, more than other well-established markers of LA function and pressure estimation, we still hold a conservative view since these measurements are components of the LA stiffness calculation equation. This limitation could be resolved by a pure statistical comparison of these parameters, but this is very complex in patients enrolled purely on clinical grounds.

**Clinical implications:** Our findings show that increased LA stiffness was associated with increased cardiac events. It is likely that increased LA stiffness could have contributed to the impairment of quality of life through perpetual increase in LA pressure and its retrograde transmission to the pulmonary circulation [31]. Therefore, in patients in whom spectral Doppler markers of raised LA pressure are not conclusive, LA stiffness measurements could add important information about the need for optimal LA pressure lowering strategies, which if ignored are known for their impact on subendocardial circulation and arrhythmias, which could be life threatening [10,35]. Thus, our findings emphasis the important role of routine incorporation of LA function assessment using various available Doppler echocardiographic techniques in daily examination of patients with HFrEF and HFmrEF. 

## 5. Conclusions

In this cohort of ambulatory HFrEF and HFmrEF patients, LASt proved the most powerful predictor of clinical outcome, particularly in patients with clear Doppler signs of raises LV filling pressures.

## Figures and Tables

**Figure 1 jcm-09-01244-f001:**
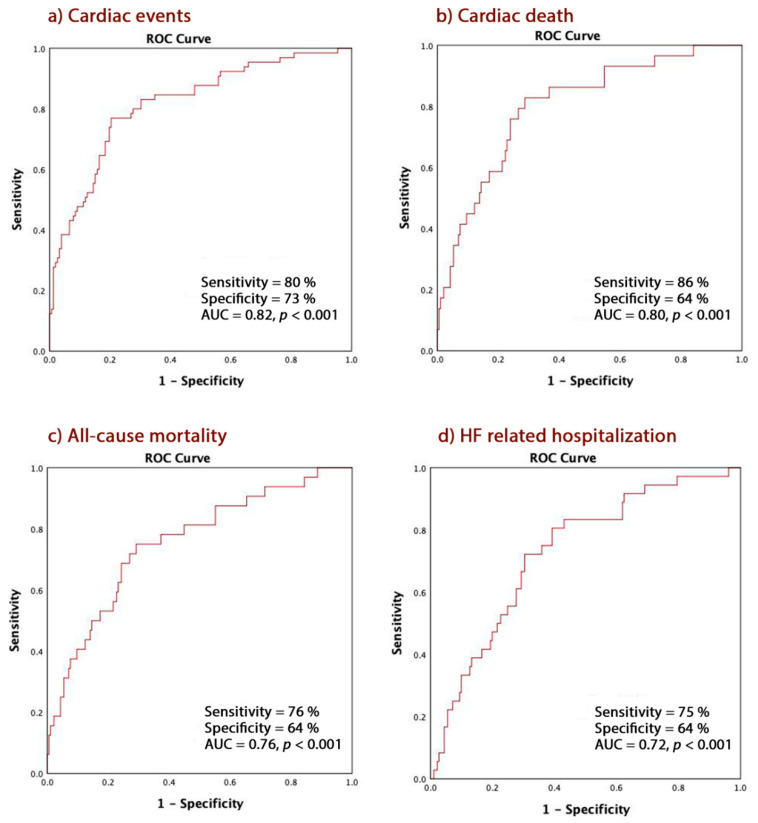
LA stiffness ≥0.76% in predicting. (**a**) Cardiac events; (**b**) Cardiac death; (**c**) All-cause mortality; (**d**) HF-related hospitalization. HF: heart failure; AUC: area under curve; ROC: receiver operator characteristic.

**Figure 2 jcm-09-01244-f002:**
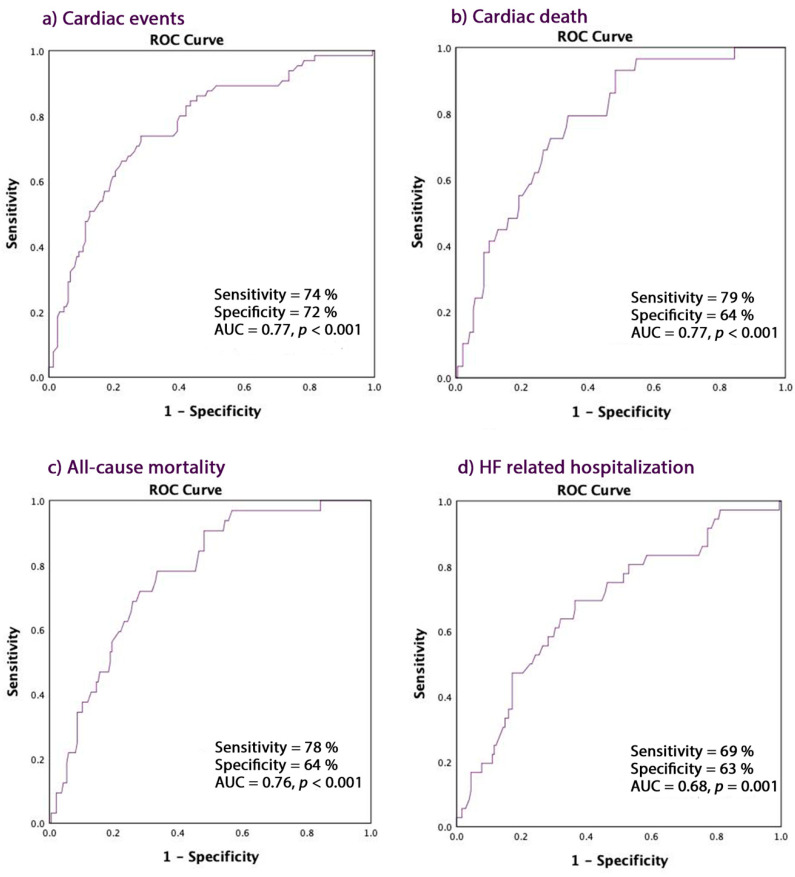
PALS ≤ 16 in predicting. (**a**) Cardiac events; (**b**) Cardiac death; (**c**) All-cause mortality; (**d**) HF-related hospitalization. PALS: peak atrial longitudinal strain.

**Figure 3 jcm-09-01244-f003:**
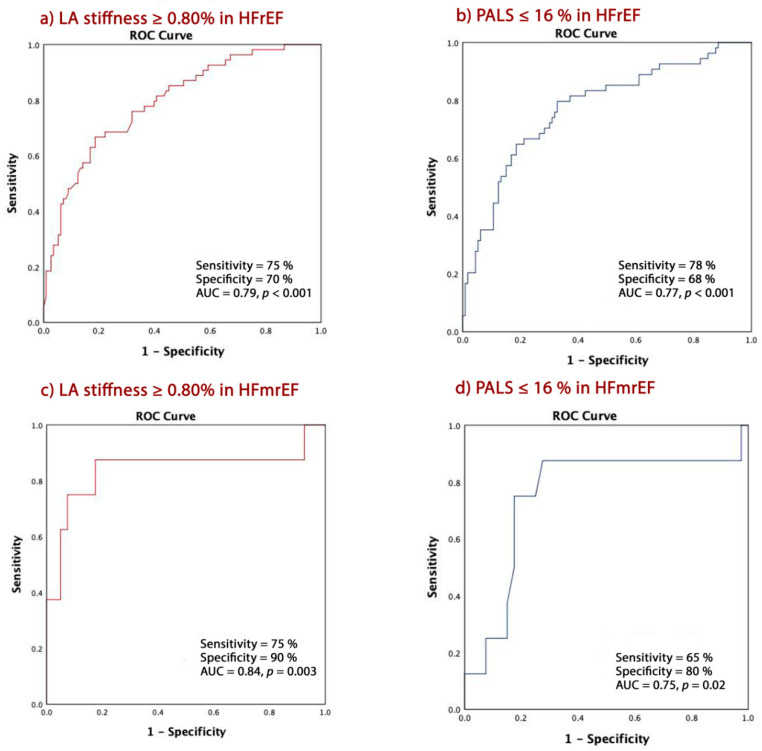
LA stiffness and PALS in predicting cardiac events in patients with HFrEF and HFmrEF; (**a**) LA stiffness ≥0.76% in cardiac events in HFrEF; (**b**) PALS ≤ 16 in predicting cardiac events in HFrEF patients; (**c**) LA stiffness ≥0.76% in cardiac events in HFmrEF; (**d**) PALS ≤ 16 in predicting cardiac events in HFmrEF patients.

**Figure 4 jcm-09-01244-f004:**
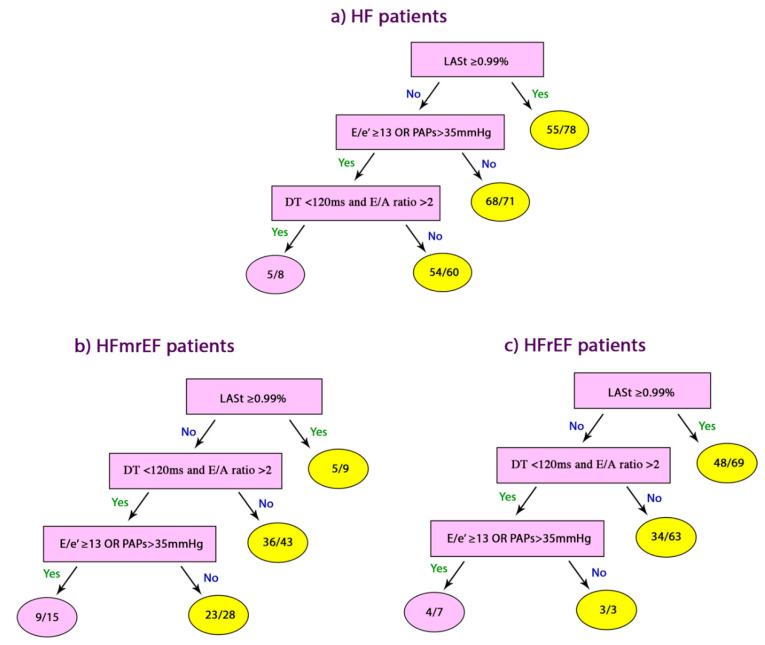
Echo predictors of CE based on CART analysis. (**a**) All HF patients; (**b**) HFmrEF patients; (**c**) HFrEF patients; CE: cardiac events; CART: Classification and Regression Tree Analyses.

**Table 1 jcm-09-01244-t001:** Baseline clinical, demographic and echocardiographic characteristics of the study groups.

Variable	CE− (*n* = 152)	CE+ (*n* = 63)	*p* Value
**Clinical data**			
Age (years)	66 ± 11	67.4 ± 11	0.34
Sex (female, *n*, %)	39 (25.6)	14 (21.5)	0.21
NYHA class	1.72 ± 0.7	2.22 ± 0.6	<0.001
AH (*n*, %)	55 (36.2)	33 (50.7)	0.01
DM (*n*, %)	40 (26.3)	28 (43.1)	0.01
CAD (*n*, %)	64 (42.1)	42 (64.6)	0.01
CRF (*n*, %)	34 (22.3)	23 (35.3)	0.001
Loop diuretics (*n*, %)	111 (73.2)	62 (95.3)	<0.0001
Beta-blockers (*n*, %)	134 (88.1)	54 (83.1)	0.12
ACE or ARBs (*n*, %)	134 (88.1)	53 (81.5)	0.08
BNP pg/mL	402 ± 465	659 ± 574	0.001
BNP > 125 (*n*, %)	104 (68.4)	55 (84.6)	0.001
**Echocardiographic data**			
LV EDD (cm)	6.0 ± 0.6	6.3 ± 0.7	0.01
LV ESD (cm)	4.7 ± 0.8	5.1 ± 0.7	<0.001
EDV (mL)	187 ± 51	207 ± 59	0.01
ESV (mL)	124 ± 43	143 ± 57	0.009
LV EF (%)	34 ± 6.1	30 ± 7.1	0.001
E wave (cm/s)	69.3 ± 23	80.4 ± 20	0.002
A wave (cm/s)	85.1 ± 47	71.3 ± 29	0.24
E/A ratio	1.03 ± 0.8	1.30 ± 0.6	0.048
E/A>2 (*n*, %)	17 (11.1)	16 (24.6)	0.004
E wave DT (ms)	190 ± 64	163 ± 58	0.01
E/e’ ratio	12.3 ± 4.7	16.0 ± 6.4	<0.001
E/e’ ratio > 15 (*n*, %)	39 (25.6)	32 (49.2)	<0.001
TAPSE (cm)	19.9 ± 3.3	18.1 ± 3.7	0.001
PAP mmHg	33.8 ± 9.2	43.0 ± 10	<0.001
Mitral regurgitation (*n*, %)	55 (35.2)	35 (60.3)	<0.001
LA diameter (cm)	4.4 ± 0.8	4.9 ± 0.7	0.001
LAVI max (mL/m^2^)	34.2 ± 11	43.4 ± 15	<0.001
LAVI min (mL/m^2^)	20.9 ± 11	31.9 ± 14	<0.001
PALS (%)	22.3 ± 9.3	12.9 ± 6.7	<0.001
LASt (%)	0.68 ± 0.4	1.6 ± 1.1	<0.001

CE: cardiac events; (−): no cardiac events; (+):cardiac events; NYHA: New York Heart Association; AH: arterial hypertension; DM: diabetes mellitus; CAD: coronary artery disease; CRF: chronic renal failure; ACE: Angiotensin-converting enzyme; ARBs: angiotensin II receptor blockers; LV: left ventricle; EDD: end-diastolic dimension; ESD: end-systolic dimension; EDV: end-diastolic volume; ESV: end systolic volume; TAPSE: tricuspid annular plane systolic excursion; PAP: pulmonary artery systolic pressure; LA: left atrium; LAVI max/min: Left atrial maximal/minimal volume/indexed.

**Table 2 jcm-09-01244-t002:** Predictors of cardiac events.

Variable	Univariate Predictors Odds Ratio (95% CI)	*p* Value	Multivariate Predictors Odds Ratio (95% CI)	*p* Value
Age	1.013 (0.978 to 1.039)	0.33		
Gender	1.2578 (0.628 to 2.518)	0.51		
NYHA class	2.638 (1.712 to 4.065)	<0.001	1.394 (0.791 to 2.456)	0.25
LV EDD	2.407 (1.538 to 3.767)	0.001		
LV ESD	1.066 (1.027 to 1.106)	0.002		
LV EDV	1.007 (1.001 to 1.012)	0.011		
LV ESV	1.008 (1.002 to 1.014)	0.010		
LV EF	0.915 (0.876 to 0.957)	<0.001	0.980 (0.923 to 1.040)	0.50
E/A ratio	1.329 (0.980 to 1.803)	0.066		
E/e’ ratio	1.128 (1.065 to 1.194)	<0.001		
E/e’ >15	2.810 (1.538 to 5.158)	0.001	0.292 (0.099 to 0.859)	0.02
TAPSE	0.850 (0.774 to 0.934)	0.001		
PAP	1. 096 (1.059 to 1.134)	<0.001	1.050 (1.009 to 1.094)	0.01
LA diameter	1.124 (1.068 to 1.184)	0.001	1.027 (0.953 to 1.107)	0.48
LAVI max	1.052 (1.028 to 1.077)	<0.001	1.000 (0.916 to 1.097)	0.69
LAVI min	1.066 (1.040 to 1.093)	<0.001	1.019 (0.928 to 1.118)	0.59
PALS	0.843 (0.798 to 0.891)	<0.001	0.932 (0.873 to 0.994)	0.02
LASt	3.122 (2.098 to 4.476)	<0.001	3.781 (1.144 to 5.122)	0.001

New York Heart Association LV: left ventricle; EDD: end-diastolic dimension; ESD: end-systolic dimension; TAPSE: tricuspid annular plane systolic excursion; LA: left atrial; LAVI max: left atrial volume index; LAVI min: left atrial volume index.

**Table 3 jcm-09-01244-t003:** Diagnostic accuracy of echocardiographic parameters in predicting CE in HF patients.

Predictors	Sensitivity	Specificity	PPV	NPV	Accuracy
**All patients**					
LASt ≥ 99%	70 (59–80)	91 (87–97)	85 (76–92)	84 (80–88)	85 (74–92)
E/e’ ≥ 13 OR PAPs > 35 mmHg	66 (53–77)	96 (88–99)	93 (82–98)	81 (74–87)	82 (79–89)
DT <120 ms and E/A >2	52 (18–84)	90 (79–96)	42 (22–65)	85 (74–92)	81 (73–87)
**HFmrEF**					
LASt ≥ 99%	57 (28–86)	84 (70–93)	45 (25–66)	90 (82–95)	79 (67–89)
E/e’ ≥ 13 OR PAPs > 35 mmHg	65 (36–87)	82 (63–93)	65 (44–81)	82 (69–90)	76 (61–88)
DT <120 ms and E/A >2	52 (18–93)	84 (69–93)	25 (12–48)	95 (87–98)	81 (67–91)
**HFrEF**					
LASt ≥ 99%	86 (75–91)	78 (67–85)	70 (61–77)	93 (87–97)	82 (75–88)
E/e’ ≥ 13 OR PAPs > 35 mmHg	58 (20–88)	100 (29–100)	100 (90–100)	50 (30–70)	70 (35–93)
DT <120 ms and E/A >2	71 (42–91)	50 (37–60)	37 (20–34)	88 (75–94)	56 (42–65)

LASt: Left atrial stiffens; LV: left ventricle; PAPs: pulmonary artery systolic pressure HFrEF; Heart failure reduced ejection fraction; HFmrEF: Heart failure mid-range ejection fraction; PPV: positive predictive value; NPV: negative predictive value.

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
