# Peer review of "Speckle Tracking-Derived Left Atrial Stiffness Predicts Clinical Outcome in Heart Failure Patients with Reduced to Mid-Range Ejection Fraction"

_jcm, 2020, doi:10.3390/jcm9051244_

Round 1
Reviewer 1 Report
The topic of the presented manuscript is very interesting and can be useful from daily practical point of view. For many years only LV size and function have been assessed as the main prognostic nonivasive markers in pts with heart failure. It is well known, that progressive and prolonged left atrial pressure overload leads to left atrial dilatation and wall fibrosis with reduced LA function and it is associated not only with the severity of symptoms, but i san important prognostic marker in patients with heart failure. Noninvasively achieved with two-dimensional speckle tracking echocardiography peak longitudinal left atrial strain (PLAS) was found to accurately estimate the degree of atrial dysfunction. The peak left atrial strain as a single measure for the non-invasive assessment of left ventricular filling pressures is strongly associated with development of dyspnea, decompensation and prognosis in patients with HF (Carluccio, E at all. (Circ Cardiovasc Imaging. 2018;11). Invasive study showed, that the most adequate parameter of LA dysfunction is increased atrial stiffness. Actually available newest and non-invasively estimated using Doppler echocardiographic parameter of LA stiffnes (obtained by dividing E/e’ Doppler ratio by PLAS) is an accurate in identifying patients with diastolic heart failure. In recent paper authors evaluated the relationship between left atrial strain and non-invasive stiffness indexes and cardiac events in patients with heart failure and LV EF <50% during a median follow up 41+- 34 months. This is the first manuscript exploring the influence of non-invasive left atrial stiffnes on clinical course and prognosis. In my opinion moderate/small mitral regurgitation and common of chronic renal failure can influence the results and should be included in the strategy in the future, but it is very promising work.
But there is some notice
Line 38 PALS - there is only abbreviation in the abstract.
Line 139 there is global peak LA strain ( PLAS) but it should be rather peak longitudinal atrial /systolic/ strain (PALS) – there is no picture , it should be explained in the method’s section.
Line 173 pts with moderate mitral regurgitation are disscused ( how many pts had moderate MR? Is there any relationship between MR and LA stiffness)
Line 179 there is : pts with CE.. , had higher BNP level (p=0,001) – in table 1 there is no BNP level, but only number pts or % of pts with BNP >125
Line 193 add; Mitral regurgitation was more frequent in CE group vs no CE.
Line 204 I’m not statisticians, but collinearity between these measurements based on VIF<10 ( but >5) for continuous data appears high,
Line 214 is p<0.001, should be p=0.001 ? ( Figure)
Line 220 for MACE ?? New abbreviation?
Line 221 p<0.001 in the text and figure 3b p< 0,01?
Line 243 LA stiffness was related to raised LA pressure… – this sentence is not the result of manuscript
Line 454 HF or CV hospitalization?
Author Response
Reviewer 1
The topic of the presented manuscript is very interesting and can be useful from daily practical point of view. For many years only LV size and function have been assessed as the main prognostic noninvasive markers in pts with heart failure. It is well known, that progressive and prolonged left atrial pressure overload leads to left atrial dilatation and wall fibrosis with reduced LA function and it is associated not only with the severity of symptoms, but is an important prognostic marker in patients with heart failure. Noninvasively achieved with two-dimensional speckle tracking echocardiography peak longitudinal left atrial strain (PLAS) was found to accurately estimate the degree of atrial dysfunction. The peak left atrial strain as a single measure for the non-invasive assessment of left ventricular filling pressures is strongly associated with development of dyspnea, decompensation and prognosis in patients with HF (Carluccio, E at all. (Circ Cardiovasc Imaging. 2018;11).
Invasive study showed, that the most adequate parameter of LA dysfunction is increased atrial stiffness. Actually, available newest and non-invasively estimated using Doppler echocardiographic parameter of LA stiffness (obtained by dividing E/e’ Doppler ratio by PLAS) is an accurate in identifying patients with diastolic heart failure. In recent paper authors evaluated the relationship between left atrial strain and non-invasive stiffness indexes and cardiac events in patients with heart failure and LV EF <50% during a median follow up 41+- 34 months. This is the first manuscript exploring the influence of non-invasive left atrial stiffness on clinical course and prognosis. In my opinion moderate/small mitral regurgitation and common of chronic renal failure can influence the results and should be included in the strategy in the future, but it is very promising work. But there is some notice:
Comments:
- Line 38 PALS - there is only abbreviation in the abstract.
Response: Thank you. We have now corrected this.
- Line 139 there is global peak LA strain (PLAS) but it should be rather peak longitudinal atrial /systolic/ strain (PALS) – there is no picture, it should be explained in the method’s section.
Response: Thank you for your suggestion. We have added a figure that explains the measurements in the revised manuscript.
- Line 173 pts with moderate mitral regurgitation are discussed (how many pts had moderate MR? Is there any relationship between MR and LA stiffness)?
Response: Thank you for your suggestion. Our patients had insignificant (less than moderate) mitral regurgitation and the relationship between MR and LASt was poor (r=0.22, p=0.002).
- Line 179 there is: pts with CE had higher BNP level (p=0,001) – in table 1 there is no BNP level, but only number pts or % of pts with BNP >125.
Response: Thank you. We have added the results in the table 1.
- Line 193 add; Mitral regurgitation was more frequent in CE group vs no CE.
Response: Thank you. We have added this in the revised manuscript.
- Line 204 I’m not statisticians, but collinearity between these measurements based on VIF<10 (but >5) for continuous data appears high,
Response: Thank you. We referred to the many statistician recommendations as minimum tolerance of VIF until 10 (<10 no collinearity, 10 to 30 indicate a possible problem and >30 suggesting a very serious problem), but in our stats no variable had a VIF value >5.
- Line 214 is p<0.001, should be p=0.001? (Figure)
Response: Thank you. We have now corrected this mistake.
- Line 220 for MACE?? New abbreviation?
Response: Thank you. We have now corrected this.
- Line 221 p<0.001 in the text and figure 3b p< 0,01?
Response: Thank you. We have now corrected the figure 3b.
- Line 243 LA stiffness was related to raised LA pressure… – this sentence is not the result of manuscript
Response: Our new stats showed now that LASt was related to markers of raised LA pressure. This has now been clarified in Line 249.
- Line 454 HF or CV hospitalization?
Response: Thank you. We have corrected this in the revised manuscript.
Reviewer 2 Report
Bytyci I, et al. revealed that noninvasive estimation of LA stiffness by speckle tracking can be a powerful prediction for clinical outcome in HF patients with reduced to mid-ranged EF.
LV end-systolic volume has been established to be an independent prognostic factor in patients with HFrEF. Moreover, it’s also clear that systolic LA strain and the reservoir function turn to be abnormal in patients with systolic function. Calculating LA stiffness (LASt) index would be still a relatively new technique to estimate LA compliance as a reservoir function. It thereby could be potentially intriguing to examine if the calculation can be the most powerful predictor of clinical outcome in the cohort of ambulatory HFrEF patients.
However, the trigger of this study approach including HFmrEF group is skeptical and I would have to say it's difficult to agree with that the results have clinical implications for sure because the therapeutic strategy is known to be totally different to HFrEF so far. It goes too far that LASt reflects LA cavity stiffness only. The superiority of LASt should be also carefully argued compared to PAP, another independent predictor of CE in the current multivariate analysis.
Author Response
Reviewer 2
Bytyci I, et al. revealed that noninvasive estimation of LA stiffness by speckle tracking can be a powerful prediction for clinical outcome in HF patients with reduced to mid-ranged EF.
LV end-systolic volume has been established to be an independent prognostic factor in patients with HFrEF. Moreover, it’s also clear that systolic LA strain and the reservoir function turn to be abnormal in patients with systolic function. Calculating LA stiffness (LASt) index would be still a relatively new technique to estimate LA compliance as a reservoir function. It thereby could be potentially intriguing to examine if the calculation can be the most powerful predictor of clinical outcome in the cohort of ambulatory HFrEF patients.
However, the trigger of this study approach including HFmrEF group is skeptical and I would have to say it's difficult to agree with that the results have clinical implications for sure because the therapeutic strategy is known to be totally different to HFrEF so far. It goes too far that LASt reflects LA cavity stiffness only. The superiority of LASt should be also carefully argued compared to PAP, another independent predictor of CE in the current multivariate analysis.
Response: Thank you for your suggestion. We have tested the power of PAP in predicting CE and added the results.
Reviewer 3 Report
In this submission, the author evaluated the relationship between left atrial stiffness (LASt), as an important marker of cardiac dysfunction in patients with heart failure (HF), and cardiac events (CE), including HF hospitalization and cardiac death, in HF patients with reduced to mid-range ejection fraction. From the result, they proved that LASt is the most powerful predictor of clinical outcomes. However, there are several issues that need to be addressed.
- The authors evaluated the relationship between left atrial stiffness (LASt) and cardiac event (CE) in HF patients with reduced to mid-range ejection fraction. For CE, they included HF hospitalization and cardiac death, and they also mentioned that they observed primary endpoints (cardiac events, combined death and hospitalization for worsening HF) and second endpoints (all-cause mortality, cardiac death and hospitalization). First of all, HF hospitalization as an endpoint may lead to some bias because some patients may be hospitalized for, for example, arrythmias, which is a HF related hospitalization. So, “HF related hospitalization” might be a better endpoint. Secondly, the first and second endpoints have overlap to some extent. And it is unnecessary to separate them for the author did not analyze them separately. What’s more, the second endpoints were inappropriate.
- The author used univariate analysis and multivariate logistic regression to evaluate the predictive power of LASt for CE. In multivariate logistic regression, the author included both E/e’ ratio and LASt. Although collinearity analysis had been done, LASt were calculated by E/e’/PALS. Could the author show the details of the collinearity about that?
- The author used LASt≥0.76% and PALS ≤16% as a cutoff value. More details should be given about how they chose the cutoff value, and why it is reasonable.
- In table 3, the author mentioned “E/e’>15 vs. E/e’<15”, does it mean they separated all patients into two groups according to whether E/e’>15? And how about patients with E/e’=15. And what is more important is, the author compared the patients with E/e’>15 + increased LASt and with E/e’<15 and decreased LASt. According to my understanding, those two groups represent the most severe and the mildest situations, respectively, so how about the patients in between, i.e., E/e’>15 + no changed/decreased LASt or E/e’<15 and on change or increased LASt? The comparison only between those two groups may lead to bias for the results. What is the definition of increase and decrease? Same problems for LV filling pattern and LASt also in table 3.
- The author mentioned the exclusion criteria. Was age under their consideration?
- What was the method did the author use when analyzed the NYHA class? T-test should not been used for comparing these ranked data.
- In conclusion, the author mentioned that “in this cohort of patients with HFrEF”. How about the patients with HFmrEF? This conclusion is different from the conclusion in the abstract.
Minor comments:
- There is some confusion about the patient’s number. The text stated 65 (30%) patients had CE and the remaining 150 (69.7%) had no CE. How about the left 0.3%? And in table 1, it stated 63 patients had CE and the remaining 152 had no CE.
- In line 63, the “physiology” should be “pathology”.
- “Biochemical test” is not appropriate. “Blood test” should be better because the author included blood count, which is not a biochemical test.
- MACE only showed up once in the whole paper. Please make sure if it is correct or not.
- The number of digits after the decimal point needs to be unified in the table, especially for the same parameter in different groups, like age and E/A ratio.
- Please give the full name before using the abbreviation, like PALS (in the abstract), ESC, A wave, and so on. And in methods, echocardiography examination, the meaning of E wave, A wave, e’ wave should be explained firstly.
- The Author mentioned that CE patients had a higher BNP level, but in the table, only BNP>125 (n, %) have been list. So, it is not appropriate to the state like that.
Author Response
Reviewer 3
In this submission, the author evaluated the relationship between left atrial stiffness (LASt), as an important marker of cardiac dysfunction in patients with heart failure (HF), and cardiac events (CE), including HF hospitalization and cardiac death, in HF patients with reduced to mid-range ejection fraction. From the result, they proved that LASt is the most powerful predictor of clinical outcomes. However, there are several issues that need to be addressed.
1.The authors evaluated the relationship between left atrial stiffness (LASt) and cardiac event (CE) in HF patients with reduced to mid-range ejection fraction. For CE, they included HF hospitalization and cardiac death, and they also mentioned that they observed primary endpoints (cardiac events, combined death and hospitalization for worsening HF) and second endpoints (all-cause mortality, cardiac death and hospitalization). First of all, HF hospitalization as an endpoint may lead to some bias because some patients may be hospitalized for, for example, arrythmias, which is a HF related hospitalization. So, “HF related hospitalization” might be a better endpoint. Secondly, the first and second endpoints have overlap to some extent. And it is unnecessary to separate them for the author did not analyze them separately. What’s more, the second endpoints were inappropriate.
Response: Thank you for your suggestion. We have now revised the discussion along the lines of your comments.
2.The author used univariate analysis and multivariate logistic regression to evaluate the predictive power of LASt for CE. In multivariate logistic regression, the author included both E/e’ ratio and LASt. Although collinearity analysis had been done, LASt were calculated by E/e’/PALS. Could the author show the details of the collinearity about that?
Response: Thank you. We tested the collinearity of the results and stated it in the methods section. The collinearity was not met.
3.The author used LASt≥0.76% and PALS ≤16% as a cutoff value. More details should be given about how they chose the cutoff value, and why it is reasonable.
Response: Thank you. We did not use those cut offs, but they were the results of our analysis of variables that could predict CE
4.In table 3, the author mentioned “E/e’>15 vs. E/e’<15”, does it mean they separated all patients into two groups according to whether E/e’>15? And how about patients with E/e’=15. And what is more important is, the author compared the patients with E/e’>15 + increased LASt and with E/e’<15 and decreased LASt. According to my understanding, those two groups represent the most severe and the mildest situations, respectively, so how about the patients in between, i.e., E/e’>15 + no changed/decreased LASt or E/e’<15 and on change or increased LASt? The comparison only between those two groups may lead to bias for the results. What is the definition of increase and decrease? Same problems for LV filling pattern and LASt also in table 3.
Response: Thank you for your comments! According to your suggestion, we have tested the cardiac events in different groups and added the results in manuscript.
5.The author mentioned the exclusion criteria. Was age under their consideration?
Response: Thank you. We excluded the age <18 years. We have now added this clarification to the methods section.
6.In conclusion, the author mentioned that “in this cohort of patients with HFrEF”. How about the patients with HFmrEF? This conclusion is different from the conclusion in the abstract.
Response: Thank you. We have now corrected this mistake.
Minor comments:
- There is some confusion about the patient’s number. The text stated 65 (30%) patients had CE and the remaining 150 (69.7%) had no CE. How about the left 0.3%? And in table 1, it stated 63 patients had CE and the remaining 152 had no CE.
Response: Thank you. We have now corrected this mistake.
- In line 63, the “physiology” should be “pathology”.
- Response: Thank you. Now we have corrected this.
- “Biochemical test” is not appropriate. “Blood test” should be better because the author included blood count, which is not a biochemical test.
Response: Thank you. Now we have corrected this.
- MACE only showed up once in the whole paper. Please make sure if it is correct or not.
Response: Thank you. We have corrected this in the revised manuscript.
- The number of digits after the decimal point needs to be unified in the table, especially for the same parameter in different groups, like age and E/A ratio.
Response: Thank you for the suggestion. We have now corrected this error.
- Please give the full name before using the abbreviation, like PALS (in the abstract), ESC, A wave, and so on. And in methods, echocardiography examination, the meaning of E wave, A wave, e’ wave should be explained firstly.
Response: Thank you for the suggestion. We have now corrected this in the revised manuscript.
- The Author mentioned that CE patients had a higher BNP level, but in the table, only BNP>125 (n, %) have been list. So, it is not appropriate to the state like that.
Response: Thank you for the suggestion. We have now corrected this error.
Round 2
Reviewer 2 Report
As pointing out the last review, the background of this study’s aim and the clinical significance remain uncertain. If the author would like to argue the current subject regarding echocardiographic techniques, It would be first required to evaluate the relationship between non-invasive LA stiffness and cardiac events after dividing into two groups such as patients with HFrEF or HFmrEF, respectively. Before starting the author's topic in the combined groups, making sure those results would help to tell the quality of current study.
Author Response
Reviewer 1
As pointing out the last review, the background of this study’s aim and the clinical significance remain uncertain. If the author would like to argue the current subject regarding echocardiographic techniques. It would be first required to evaluate the relationship between non-invasive LA stiffness and cardiac events after dividing into two groups such as patients with HFrEF or HFmrEF, respectively. Before starting the author's topic in the combined groups, making sure those results would help to tell the quality of current study.
Response: Thank you. We have compared the relationship between LA stiffness between the two subgroups of patients with HFrEF and HFmrEF and found no difference
Reviewer 3 Report
The authors partially addressed my comments, however, two critical questions remain:
1. Please report the collinearity test results to the reviewer.
2. For the LASt and PALS cutoff values, the response is not acceptable. You either based on normal values or did multiple trials of ROC curve testing to get the best values. They are essentially cut off values.
3. For Table 3, statistically, it was a wrong way to go. First of all, there was no explanation why 15 was used for E/e’ and then what is defined as LASt increased or decreased. All those values were arbitrary. Please scientifically justify your cut-off values. At last, to get a real comparison of the predicting power of the combinations of E/e’>=15 or <15 with LASt increases or decreases, logistic regression must be done first to really determine if the combination is better in predicting power. After the logistic regression, then pair-wise comparison with the control group by OR analysis could be done to make the point.
Author Response
Reviewer 2
The authors partially addressed my comments; however, two critical questions remain:
- Please report the collinearity test results to the reviewer.
Response: Thank you. We have now reported the collinearity to reviewers and Editors.
- For the LASt and PALS cutoff values, the response is not acceptable. You either based on normal values or did multiple trials of ROC curve testing to get the best values. They are essentially cut off values.
Response: Thank you. We have now tested the cut off value of predictors based on ROC analysis and also tested them with CART analysis.
- For Table 3, statistically, it was a wrong way to go. First of all, there was no explanation why 15 was used for E/e’ and then what is defined as LASt increased or decreased. All those values were arbitrary. Please scientifically justify your cut-off values. At last, to get a real comparison of the predicting power of the combinations of E/e’>=15 or <15 with LASt increases or decreases, logistic regression must be done first to really determine if the combination is better in predicting power. After the logistic regression, then pair-wise comparison with the control group by OR analysis could be done to make the point.
Response: Thank you for the suggestion. We have now changed the combination of predictors based on classification and regression tree analysis. Finally, we have repeated the logistic regression analysis for different combinations followed by pair wise comparison with the control group, as suggested.